# Bioengineered Silkworm for Producing Cocoons with High Fibroin Content for Regenerated Fibroin Biomaterial-Based Applications

**DOI:** 10.3390/ijms23137433

**Published:** 2022-07-04

**Authors:** Mana Yamano, Ryoko Hirose, Ping Ying Lye, Keiko Takaki, Rina Maruta, Mervyn Wing On Liew, Shinichi Sakurai, Hajime Mori, Eiji Kotani

**Affiliations:** 1Department of Applied Biology, Kyoto Institute of Technology, Sakyo-ku, Kyoto 606-8585, Japan; manakichy.y@gmail.com (M.Y.); hrro0621@gmail.com (R.H.); ying5434@hotmail.com (P.Y.L.); ktakaki@kit.ac.jp (K.T.); rina.maruta07@gmail.com (R.M.); silk7776@gmail.com (H.M.); 2Biomedical Research Center, Kyoto Institute of Technology, Sakyo-ku, Kyoto 606-8585, Japan; 3Institute for Research in Molecular Medicine, Universiti Sains Malaysia, Minden, Penang 11800, Malaysia; mervynliew@yahoo.com; 4Department of Biobased Materials Science, Kyoto Institute of Technology, Sakyo-ku, Kyoto 606-8585, Japan; shin1sakuri@gmail.com; 5Department of Chemical Engineering, Indian Institute of Technology Guwahati, Guwahati 781039, Assam, India

**Keywords:** *Bombyx mori*, fibroin, biomaterial, bone regeneration

## Abstract

Silk fibroin exhibits high biocompatibility and biodegradability, making it a versatile biomaterial for medical applications. However, contaminated silkworm-derived substances in remnant sericin from the filature and degumming process can result in undesired immune reactions and silk allergy, limiting the widespread use of fibroin. Here, we established transgenic silkworms with modified middle silk glands, in which sericin expression was repressed by the ectopic expression of cabbage butterfly-derived cytotoxin pierisin-1A, to produce cocoons composed solely of fibroin. Intact, nondegraded fibroin can be prepared from the transgenic cocoons without the need for sericin removal by the filature and degumming steps that cause fibroin degradation. A wide-angle X-ray diffraction analysis revealed low crystallinity in the transgenic cocoons. However, nondegraded fibroin obtained from transgenic cocoons enabled the formation of fibroin sponges with varying densities by using 1–5% (*v*/*v*) alcohol. The effective chondrogenic differentiation of ATDC5 cells was induced following their cultivation on substrates coated with intact fibroin. Our results showed that intact, allergen-free fibroin can be obtained from transgenic cocoons without the need for sericin removal, providing a method to produce fibroin-based materials with high biocompatibility for biomedical uses.

## 1. Introduction

Cocoon silk produced by the silkworm *Bombyx mori* is composed of an insoluble inner fibroin layer surrounded by a hydrophilic outer sericin layer [1,2,3,4,5]. Sericin and fibroin are expressed in the middle silk gland (MSG) and posterior silk glands (PSGs), respectively, during the late fifth larval stage upon cocoon formation. Fibroin, which consists mainly of fibroins H and L, together with a small amount of p25 fibrohexamarin, accounts for 75% of the total cocoon proteins. The remaining 25% cocoon proteins consist mainly of sericin (sericin 1 and 3, hereafter abbreviated as Ser1 and Ser3, respectively). When fibroin is processed into silk threads, sericin is removed by filature, which involves boiling the cocoon, followed by degumming in an alkaline solution [4,5].

Fibroin-based materials are useful for biomedical applications, such as in fine sutures [6], electrospun nanofiber film-based dressings [7], hydrogels for drug delivery and tissue engineering [8,9], biodegradable carriers for cell growth factors [10,11], and sponge scaffolds for cell growth [12,13,14]. Fibroin-based biomaterials, which are biocompatible, biodegradable, and have low immunostimulatory properties, have been used for revascularization [15], bone tissue regeneration [16], and cutaneous wound healing [17,18]. Our previous study showed that fibroin powders incorporated with protein microcrystal-encapsulated cytokines processed from transgenic silk glands effectively controlled NIH-3T3 proliferation [10] and facilitated the formation of a three-dimensional (3D) epidermis model [11]. These examples highlight the potential of fibroin-based biomaterials that can be generated through the bioengineering of silkworms for biomedical applications, particularly for tissue engineering.

The Pierisin-1A (P1A) protein (approximately 100 kDa) is a member of the pierisin family and is produced by the cabbage butterfly, *Pieris rapae*. P1A is a homolog of pierisin-1 that induces rapid apoptosis in numerous mammalian cells [19,20,21,22,23,24]. P1A consists of an N-terminal region with an ADP-ribosyltransferase domain (~269 amino acids) that catalyzes the transfer of the ADP-ribose moiety of NAD to the 2′-deoxyguanosine residues of the DNA [19,20], along with a putative domain at the C-terminus that mediates its binding to cell membrane receptors and uptake by target cells [22,23,24]. We previously demonstrated that the expression of the P1A catalytic domain (P1A269), which has a lower enzymatic activity than other pierisin family proteins, results in the repression of the reporter gene expressed in an insect cell line without causing apoptosis [20]. More importantly, the expression of P1A269 in silkworm PSGs resulted in the repression of fibroin synthesis, leading to the formation of fibroin-free, sericin cocoons [20].

Based on our previous findings demonstrating the ectopic expression of P1A269 that represses multiple genes involved in fibroin synthesis, we predicted that multiple genes could be repressed in other silkworm tissues, and tissue-specific dysfunction could be induced by the insertion of the P1A enzyme-coding region through transgenesis [20]. The suppression of a specific gene function has been achieved using gene silencing or editing in many organisms [25,26]. Additionally, genetically manipulated model organisms with the cytotoxic protein-induced dysfunction of specific tissues will facilitate the study of various biological mechanisms.

The filature and degumming steps performed to remove the sericin layer from cocoon threads are inefficient for producing allergen-free silk threads for biomedical purposes, because contamination with silkworm-derived biological matter often occurs [27]. Furthermore, the high consumption of an alkaline solution and the generation of waste (degraded sericin) during the sericin removal process are environmental concerns. To circumvent the need for sericin removal steps and facilitate fibroin biomaterial preparation, it may be possible to use a hypothetical transgenic silkworm with modified traits that produces cocoons consisting solely of fibroin. In this study, the ectopic expression of P1A in silkworm MSG during the late larval stage resulted in the establishment of sericin-free fibroin cocoons that can be utilized to fabricate fibroin biomaterials with a regenerative capacity for cultivating mouse bone tissue model cells [28].

## 2. Results

### 2.1. Generation of Bioengineered Silkworms Producing Cocoons with High Fibroin Content

To produce novel cocoons with a high fibroin content, transgenic Ser1-P1A269 and Ser3-P1A269 silkworm lines with modified MSGs expressing the P1A catalytic domain (P1A269) [20] under the control of the Ser1 [29] and Ser3 [30] promoters, respectively, were generated using constructed donor plasmids (Figure 1A,B) (see Section 4; Appendix A). Transgene insertion into chromosome 27 of the Ser1-P1A269 line and chromosome 5 of the Ser3-P1A269 line was confirmed using inverse polymerase chain reaction (PCR). The expression of the P1A269 protein was detected in MSGs from day 5 of the fifth instar of the Ser1-P1A269 line (Appendix A) and day 6 of the fifth instar of the Ser3-P1A269 line (Appendix A). The effects of P1A269 expression on sericin production in MSGs were evaluated using quantitative reverse transcription (RT)-PCR and sodium dodecyl sulfate-polyacrylamide gel electrophoresis (SDS–PAGE). The quantitative RT-PCR (qRT-PCR) results showed that, on day 5 of the fifth instar, the Ser1 mRNA level in the MSGs from the Ser1-P1A269 line was only approximately 2.5% of that in the control, *w1-pnd* (*p* < 0.001, *n* = 5; Figure 2A), which was observed to peak (Appendix A). These results indicate that Ser1 mRNA transcription was significantly repressed in the MSGs of the Ser1-P1A269 line. The Ser1 protein [4,5], which was not detected in the SDS–PAGE analysis of cocoons produced by the Ser1-P1A269 line (Figure 2C, lane s1), was clearly observed in cocoons produced by *w1-pnd* (Figure 2C, lane w). In contrast, the Ser3 mRNA level in the MSGs of the Ser1-P1A269 line was comparable to that produced by *w1-pnd* (Appendix A) at day 6 of the fifth instar. An approximately 200-kDa band at the same molecular weight as the Ser3 protein has been previously reported [4,5] and was observed in the SDS–PAGE analysis of cocoons produced by the Ser1-P1A269 line. The presence of Ser3 was confirmed through the liquid chromatography–tandem mass spectrometry analysis of the excised gel band; a partially fragmented fibroin heavy chain (FibH) protein (data not shown) was also detected. These results suggest that Ser3 protein synthesis is not influenced by P1A269 expression driven by the Ser1 promoter. The analysis of Ser3-P1A269 MSGs revealed that, although the Ser1 protein levels were not reduced, the Ser3 mRNA levels were reduced by approximately 25.0% (*p* < 0.001, *n* = 8) compared to those in the control *w1-pnd* (Figure 2B,C, lane s3). Accordingly, the Ser3-P1A269 cocoons showed a low level of the 200-kDa band (Figure 2C, lane s3), which is the expected molecular mass of the Ser3 protein [4,5].

Cocoons produced by the Ser1/3-P1A269 line, which was established by crossing the Ser1- and Ser3-P1A269 lines, mainly contained the FibH and FibL proteins and showed low levels of Ser1 and Ser3 (Figure 2C, lane s13), as determined using SDS–PAGE. In contrast, smeared protein bands were observed in degummed *w1-pnd* cocoons (Figure 2C, lane dw). The smeared bands indicate that sericin and much of the fibroin were degraded following the boiling and alkaline treatments. Although degraded proteins were obtained from degummed *w1-pnd* cocoons, the fibrous morphology of the cocoons was retained, possibly because of the presence of thermodynamically stable β-sheet crystal structures (data not shown). The weights of non-degummed Ser1-P1A269 (41%) and Ser1/3-P1A269 (43%) cocoons were significantly reduced and were similar to that of degummed *w1-pnd* (approximately 31%) (*p* < 0.001, *n* = 5; Figure 2D). In contrast, Ser3-P1A269 cocoons showed no significant difference in weight from *w1-pnd* cocoons. These results indicate that intact, nondegraded fibroin containing a negligible amount of sericin can be readily obtained from both Ser1-P1A269 and Ser1/3-P1A269 cocoons. The similarity in the weights of Ser3-P1A269 and *w1-pnd* cocoons indicates that only a small amount of Ser3 proteins was lost from Ser3-P1A269-generated cocoons, which is consistent with the fact that Ser3 is less abundant than Ser1 in normal cocoons. Similarities in the percentages of glycine, alanine, and serine relative to the total amino acids of cocoons produced by the Ser1-P1A269 line to commercial silk thread fibroin prepared by degumming were also observed in the amino acid analysis (Appendix A). Glycine, alanine, and serine residues are well-known as the major constituents of silkworm fibroin. These results revealed that P1A269 expression significantly repressed the genes critical for sericin production in MSGs in a promoter-specific manner, which is similar to previously reported results [20].

### 2.2. Recrystallization of Intact Fibroin from the Ser1-P1A269 Cocoon to Form a 3D Sponge

To explore the efficiency of processing cocoons with a high fibroin content into regenerated materials, fibroin solutions prepared from transgenic or non-transgenic cocoons were used to prepare 3D porous fibroin sponges, as previously described [12]. As shown in Figure 2D, the difference in weight loss between the Ser1-P1A269 and Ser1/3-P1A269 cocoons compared to *w1-pnd* was inconsequential. Thus, Ser1-P1A269 cocoons were selected for further processing. First, Ser1-P1A269 cocoons were dissolved in concentrated lithium bromide (LiBr) solution, followed by dialyzation in water. Although large amounts of precipitates were observed with fibroin prepared from *w1-pnd* cocoons after dialysis, fibroin prepared from Ser1-P1A269, Ser1/Ser3-P1A269, and degummed *w1-pnd* cocoons did not contain precipitates (Appendix A). A previous study reported no precipitation when fibroin-free sericin-rich cocoons dissolved in concentrated LiBr were dialyzed against water [20]. These results indicate that fibroin interacted with sericin to form precipitates as the LiBr concentration in the cocoon solution gradually decreased. Additionally, the presence of sericin in cocoons could hinder the preparation of a pure fibroin solution.

Subsequently, the properties of 3D porous sponges prepared from fibroin solutions following overnight incubation in a methanol solution at −20 °C were investigated (see Materials and Methods, Section 4.5). The addition of 1% (*v*/*v*) methanol to fibroin solutions prepared from either non-degummed Ser1-P1A269 or degummed *w1-pnd* cocoons generated wet sponge-like structures (Figure 3A). This structure was also observed following the addition of 3% methanol (*v*/*v*) to the fibroin solutions prepared from Ser1-P1A269 cocoons (Figure 3A). However, under the same conditions, fibroin obtained from degummed *w1-pnd* cocoons showed insufficient recrystallization. The degummed *w1-pnd* cocoons, which were observed to exhibit a severe smear pattern in SDS–PAGE (Figure 2C), formed gel-like substrates with impaired robustness (Figure 3A). The addition of 5% (*v*/*v*) methanol to the fibroin solution prepared from Ser1-P1A269 led to the formation of a white sponge with a mottled surface and increased dark, gelated area (Appendix A). This result is consistent with those from a previous study showing that a higher volume of alcohol led to the formation of a sponge with high porosity and low robustness [12]. These results suggest that the volume of methanol added can be used to control the crystallization or gelation of fibroin solutions prepared from degummed *w1-pnd* cocoons. However, following the addition of 1–5% methanol (*v*/*v*) to intact and nondegraded fibroin obtained from Ser1-P1A269 cocoons, 3D sponges with varying pore sizes could be obtained.

Next, the densities of wet fibroin sponges prepared by adding 1% and 3% (*v*/*v*) methanol to *w1-pnd* and Ser1-P1A269 fibroin solutions were compared. The fibroin sponges prepared from *w1-pnd* showed approximately 12% and 11% higher transparency, respectively, compared to that of non-degummed Ser1-P1A269 fibroin prepared with a corresponding volume of methanol (*p* < 0.001, *n* = 3). These results indicate that high-density sponges can be readily formed from the intact and nondegraded fibroin of transgenic cocoons.

### 2.3. Wide-Angle X-ray Diffraction Analysis of Cocoons and Regenerated Fibroin Sponges

To estimate the degree of crystallinity, wide-angle X-ray diffraction (WAXD) measurements were performed using beamline BL-6A at the Photon Factory of the High-Energy Accelerator Research Organization (KEK) (Tsukuba, Japan). Lyophilized, non-degummed Ser1-P1A269, *w1-pnd* cocoons, and degummed Ser1-P1A269 cocoons were cut into pieces and measured using WAXD (see Materials and Methods, Section 4.6) [31,32]. A normal cocoon without degumming treatment was not available for this analysis, as it contained approximately 25% sericin in the outer layer, with more amorphous regions that would have affected the crystallinity analysis of fibroin in the inner core layer. Thus, the degummed *w1-pnd* cocoon was analyzed as a non-transgenic sample. To determine the crystallinity of the cocoon sample, we prepared amorphous fibroin by dissolving degummed *w1-pnd* cocoons in concentrated LiBr solution, followed by dialysis in large amounts of water and then lyophilization without the induction of fibroin recrystallization. The integrated value of the normalized WAXD intensity for each cocoon sample was subtracted from that of the amorphous sample (Figure 4; Appendix A), and the remainder was used to evaluate the degree of crystallinity of each cocoon (Table 1). The Ser1-P1A269 cocoon contained less crystallized fibroin compared to the degummed *w1-pnd* cocoon (Figure 4 and Table 1). Degumming marginally increased the crystallinity; however, the crystallinity of the degummed Ser1-P1A269 cocoon was still lower than that of the degummed *w1-pnd* cocoon (Table 1). The values listed in Table 1 are the results after the computational peak deconvolution (see Appendix A for the detailed procedure used for peak deconvolution). Homemade software was used, and the Lorentz function was assumed for the crystalline reflection peak and by adjusting the fraction of amorphous halo scattering. We assumed that the WAXD profile for the amorphous fibroin was included while partially maintaining the same shape of the profile, according to the amorphous component in the normalized WAXD profiles of other crystalline samples, as shown by the blue broken curves in Appendix A. Table 1 also lists the remarkably large half-width at half-maximum value (HWHM value) of the main peak for the non-degummed Ser1-P1A269 cocoon sample among the tested cocoons. The results suggest that the Ser1-P1A269 cocoon is composed of the smallest crystallinities of the fibroin—that is, the largest amount of insufficiently grown finest crystals among the tested cocoons.

Next, WAXD measurements for fibroin sponges prepared from non-degummed or degummed Ser1-P1A269 cocoons, along with the degummed *w1-pnd* cocoon, were performed under lyophilized dry conditions to evaluate the crystallinity. In contrast to the results for the cocoon samples, sponges regenerated by the addition of 1% (*v*/*v*) methanol to the samples showed a higher crystallinity than those to which 3% (*v*/*v*) methanol was added, regardless of whether they were degummed (Table 1; Appendix A), indicating that a lower volume of methanol causes the sponges to have a higher crystallinity. These results correspond well with the observation of a high degree of gel-like substances in wet fibroin sponges induced by a larger volume of methanol (Figure 3). However, changing from wet to dry conditions for the WAXD measurements appeared to facilitate fibroin crystallization in the sponges. The crystallinity of the sponges did not differ between degummed and non-degummed sponges; however, degummed fibroin was likely gelated by the addition of a higher volume of methanol, as compared to the non-degummed fibroin sponge (Figure 3), in which solidification was observed even under wet conditions. Thus, the recrystallization of fibroin to produce wet sponges would be superior when the fibroin molecules maintain intact and non-degraded structures.

### 2.4. Proliferation and Differentiation of Chondrogenic Cells on the Fibroin Material

Regenerated or fabricated fibroins, which have high biocompatibility, biodegradability, and processability, show good potential for biomedical applications [12,13,14,15,16,17,18]. To determine the effect of fibroin prepared from transgenic cocoons with a high fibroin content on cellular proliferation, mouse ATDC5 cells were used as a model cell line. Mouse ATDC5 cells show early phase chondrogenic and late-phase osteogenic differentiation, whereas cartilage and bone differentiation occur in a culture [28,33]. The proliferation of ATDC5 cells seeded on substrates coated with fibroin prepared from ethanol-treated non-degummed Ser1-P1A269 and Ser1/3-P1A269 or degummed *w1-pnd* cocoons was compared with that of conventionally prepared gelatin and noncoated substrates. Fibroin recrystallization induced by the addition of ethanol results in the formation of substrates that are structurally similar to fibroin sponges [12]. Cell viability was compared for seven days of cultivation in a proliferation medium (Figure 5A). However, increased cell viability was observed with the Ser1-P1A269 fibroin-coated substrate compared to the gelatin-coated substrate on day 7; most differences were insignificant after seeding. These results indicate that the fibroin coats did not adversely affect ATDC5 cell proliferation.

To assess the influence of the substrate on promoting bone regeneration in differentiation medium containing insulin and bone morphogenetic protein-2 (BMP-2), cartilage nodule formation was examined by measuring the sulfated glycosaminoglycan content using Alcian blue staining (Figure 5B). The levels of sulfated glycosaminoglycan on substrates coated with fibroin prepared from non-degummed Ser1-P1A269, non-degummed Ser1/3-P1A269, and degummed *w1-pnd* cocoons were approximately 1.52-, 1.49-, and 1.40-fold higher than those on the noncoated substrates, respectively (*p* < 0.001, *n* = 4). An insignificant increase in the glycosaminoglycan levels was observed in cells grown on gelatin-coated substrates (Figure 5B). Thus, substrates prepared with intact, nondegraded fibroin from transgenic cocoons greatly enhanced early-phase chondrogenic differentiation.

During the late osteogenic differentiation of ATDC5 cells, characteristic extracellular matrix mineralization during bone formation was measured using alizarin red staining (Figure 5C). Although cultivation on fibroin-coated substrates strongly promoted the early phase of chondrogenic differentiation (Figure 5B), the levels of osteogenic differentiation did not significantly differ between samples cultivated with fibroin materials and those cultivated with gelatin (Figure 5C).

## 3. Discussion

We found that the ectopic expression of the cytotoxin P1A catalytic domain with moderate DNA ADP-ribosylation activity (P1A269) repressed cocoon proteins such as Ser1 and Ser3 in nonproliferating somatic cells of MSGs (Figure 2) [19,20,24]. Although the precise underlying mechanism of the P1A function remains unclear, irreversible guanine mono-ADP ribosylation by P1A in cells may disrupt normal cellular mechanisms, such as signal transduction [34], that influence gene expression control in silkworm cells. We previously showed that DNA modifications by P1A activity did not induce apoptosis in nonproliferating PSG cells but considerably repressed FibH and FibL without significantly affecting the expression levels of the housekeeping genes [20]. Thus, we hypothesized that P1A269 could be utilized to repress sericin production in nonproliferating MSG cells.

Sericin genes are expressed in a region-specific manner (Appendix A), with high levels of Ser1 reported in the middle and posterior regions of MSG and Ser3 in the anterior region of MSG [5]. Accordingly, Ser3 expression was not considerably repressed in the MSGs of the Ser1-P1A269 line, as observed for Ser1. Similarly, Ser1 expression was not repressed by P1A activity driven by the Ser3 promoter in the Ser3-P1A269 line. Importantly, a high mortality of pupae during cocoon spinning was observed in all lines carrying the transgene Ser3-P1A269, except for the line presented in this article showing no individual lethality during silkworm development. In this Ser3-P1A269 line, the expression of Ser3 in MSGs was downregulated approximately four-fold (Figure 2B), which was minor compared to Ser1 mRNA repression (~40-fold) in the Ser1-P1A269 line (Figure 2A). Thus, the considerable repression of Ser3 by P1A269 may negatively affect silkworm development, as evidenced by the enhanced lethality of transgenic individuals. A possible explanation for this result is that Ser3 was not considerably repressed in the Ser3-P1A269 line with a low expression of P1A269 (Appendix A). Despite detectable Ser3 expression, the cocoon weights of the Ser1-P1A269 and Ser1/3-P1A269 lines showed a minimal difference. This result indicates that Ser3 accounts for only a small proportion of sericin protein in comparison to Ser1 (Figure 2D); thus, the cocoons from both the Ser1-P1A269 and Ser1/3-P1A269 lines were solely composed of fibroin.

Novel silkworm lines with attenuated sericin production will improve the understanding of the biological significance of sericin. Silk obtained from cocoons through filatures and degumming for textile manufacturing contains bundles of fibroin microfibrils that assemble into filaments. Residual sericin surrounds the fibroin microfibrils and prevents them from breaking, forming split ends similar to those in cottons [1,35]. The cocoons from both the Ser1-P1A269 and Ser1/3-P1A269 lines formed the same cocoon shape as the *w1-pnd* line, in which the fibroin fibers appeared to be glued together despite the substantial depletion of the silk protein sericin. The Ser1-P1A269 and Ser1/3-P1A269 cocoons were indistinguishable from those of *w1-pnd*, possibly because the splitting of microfibrils may not occur without sufficient Ser1 and Ser3, and the intact, nondegraded fibroin alone is sufficient to hold the fibers together.

Previous studies showed that, in addition to inducing the fiberization of fibroin during spinning, the Ser1 and Ser3 that secreted into the lumen of MSGs are critical for transporting liquid silk proteins through the tubular organ [36]. Sericin lowers the viscosity of liquid silk proteins to facilitate their transport through the long narrow ducts of the anterior silk glands [36]. Fibroin in PSGs not surrounded by sericin has a higher water retention capacity and viscosity compared to fibroin surrounded by sericin in the anterior regions of MSGs. This suggests that sericin not only dehydrates fibroin but also plays a role in decreasing the viscosity [36]. The lower silk yields, as indicated by the reduced cocoon weights of Ser1-P1A269 and Ser1/3-P1A269 (Figure 2D), suggest that a highly viscous fibroin solution is retained in the silk gland. The low crystallinity of the Ser1-P1A269 cocoons (Figure 5) supports the role of sericin in dehydrating fibroin, which facilitates effective crystallization [36,37]. Collectively, these results provide insights into the functional importance of sericin in the maturation of natural fibrous proteins.

Cocoons composed solely of fibroin enabled the preparation of an aqueous nondegraded fibroin solution without precipitation, possibly through the interaction between fibroin and sericin (Appendix A). Nondegraded fibroin, which can be recrystallized in 1–5% (*v*/*v*) methanol (Figure 3A), can be used to prepare fibroin sponges with varying pore sizes [12]. In contrast, degummed fibroin failed to crystallize in 3% (*v*/*v*) methanol (Figure 3A), indicating that degummed fibroin was degraded and in a conformational state that was unfavorable for recrystallization. Thus, regenerated fibroin can be readily prepared from Ser1-P1A269 and Ser1/3-P1A269 transgenic cocoons without performing the degumming steps. Cocoons composed solely of fibroin are not suitable as raw materials for preparing textiles with high tensile strength because of their low crystallinity (Figure 4). However, the low crystallinity of fibroin may ease the solubilization in low concentrations of chaotropic ions (i.e., LiBr), facilitating the preparation of regenerated fibroin-based biomaterials and minimizing the adverse environmental impacts associated with the degumming steps.

Intact, nondegraded fibroin obtained from Ser1-P1A269 and Ser1/3-P1A269 cocoons significantly enhanced the early chondrogenic differentiation of ATDC5 cells (Figure 5B). No induction of late osteogenic differentiation was observed (Figure 5C). These results suggest that fibroin structures enhance the proliferation of fibroblast-derived ATDC5 cells (Figure 5). Gut powder with intact fibroin structures incorporated with growth factors also enhanced the proliferation of mesenchymal cells [10]. Nondegraded fibroin may retain a random-coiled structure that is critical for the interaction with cellular surface proteins, which convey differentiation stimuli to the cellular cytoplasm. Fibroin from degummed cocoons likely loses its random-coiled structure and forms a β-sheet conformation after protein degradation. A previous study suggested that β-sheet structures weaken stem cell interactions with sericin-coated scaffolds and enhance cell–cell interactions [38].

The incomplete removal of remnant sericin can result in undesired immune reactions or silk allergies in biomedical applications, such as when fibroin is utilized as a biodegradable surgical suture [6,27]. The administration of pure sericin prepared directly from non-degummed cocoons does not stimulate IgE production or the abnormal induction of CD68+ macrophages and MPO+ neutrophils [27]. The conventional filature process involves boiling whole cocoons with the pupae inside. Therefore, the incomplete removal of silkworm-derived biological matter contaminated in the residual sericin layer during the degumming of cocoons or threads can cause allergenic reactions. The presence of contaminated silkworm-derived substances in residual sericin is of great concern for current uses such as silk thread sutures [27] and biomedical applications under development, such as scaffolds to grow artificial skin for wound healing [7,17,18] or as fine tubular materials with encapsulated cytokines for vascular regeneration [15]. Thus, new methods for producing sericin-free fibroin materials are needed.

## 4. Materials and Methods

### 4.1. Silkworm Strains and Cultured Cells

The non-diapausing silkworm strain *w1-pnd* was used to generate transgenic silkworms. The silkworm larvae were reared on an artificial diet (Kimono Brain Co., Ltd., Niigata, Japan).

ATDC5 cells [28] were cultured in 75 cm^2^ tissue culture flasks (Iwaki, Co., Ltd., Tokyo, Japan) containing the proliferation medium formulated with DMA/Ham’s F12 (1:1) medium (FUJIFILM Wako Laboratory Chemicals, Osaka, Japan), 0.36 mg/mL L(+)-glutamine, 10 µg/mL transferrin, 3.2 × 10^−8^ M sodium selenite, 5% fetal bovine serum (Thermo Fisher Scientific, Waltham, MA, USA), 100 U/mL penicillin, and 100 µg/mL streptomycin. The cell cultures were maintained in a 5% CO_2_ humidified incubator at 37 °C. Approximately 80% of the sub-confluent cells were washed with 10 mL of D-PBS(-) (FUJIFILM Wako Laboratory Chemicals) and then treated with 0.05% trypsin and 1 mM EDTA for 1 min at 37 °C, followed by the addition of 5 mL of proliferation medium. The cell suspension was centrifuged at 1000× *g* for 5 min, and 3.0 × 10^5^ cells were seeded into the same flask for continuous passage.

### 4.2. Generation of the Transgenic Silkworm Line Carrying MSG-Specific, Promoter-Driven P1A

Detailed information on vector construction and transgenic silkworm generation is provided in the Appendix A. An MSG-specific Ser1 promoter [29] was used. By modifying the vector with the transposon sequence [39], we constructed the donor vector pBacMCS[Ser1Pro-H1/P1A269/FLAG, 3xP3-EGFP] (Figure 1A) to generate the Ser1-P1A269 line expressing the P1A catalytic domain, P1A269 (2–269 amino acids of P1A fused in-frame with tags) [20], in MSGs under the control of the Ser1 promoter. Transgenic silkworms were obtained as previously described [40,41], and the homogenous line Ser1-P1A269 was established. A homogeneous transgenic line, Ser3-P1A269, which expresses P1A269 in MSGs under the control of the Ser3 promotor [30], was established using a similar method with the donor vector pBacMCS[Ser3Pro-H1/P1A269/FLAG, 3xP3-dsRed] (Figure 1B). Subsequently, Ser1-P1A269 was crossed with the Ser3-P1A line to generate Ser1/Ser3-P1A269. The resulting progeny were screened for marker traits (Ser1-P1A269 line, EGFP-positive eyes and nervous systems; Ser3-P1A269, DsRed-positive eyes) and then bred to homozygosity. The Ser1/3-P1A line was kept through more than 15 generations of sib mating and isolation using each marker trait.

### 4.3. SDS–PAGE Analysis of Proteins from Cocoons and Protein Band Analysis

To remove sericin, the *w1-pnd* cocoons were degummed by boiling in 0.05% Na_2_CO_3_ for 30 min at 95 °C, followed by washing with water and air drying. Cocoons (0.1 g) from the Ser1-P1A269, Ser3-P1A269, and nontreated *w1-pnd* and *w1-pnd^P1A269/P1A269^* lines [20] and fibroin fibers (0.1 g) of degummed *w1-pnd* cocoons were dissolved in 2 mL of 8 M LiBr for 2 h at 25 °C. The protein samples underwent 50-fold dilution in pure water, mixed with an equal volume of EzApply SDS–PAGE sample buffer (ATTO Co., Ltd., Tokyo, Japan), and heated for 5 min at 95 °C. Each protein sample (7.5 µg/lane) was electrophoresed on a 5–20% gradient precast mini gel (e-PAGEL; ATTO), followed by staining with Coomassie brilliant blue.

Protein samples were prepared from Ser1-P1A269 cocoons by dissolution in 8 M LiBr followed by dialysis with water for LiBr removal and then separated using SDS–PAGE. The gel fragment containing the approximately 200-kDa band was analyzed by liquid chromatography–tandem mass spectrometry (APRO Science Co., Ltd., Tokushima, Japan) to characterize the approximately 200-kDa proteins in the cocoon.

### 4.4. Quantitative RT-PCR Analysis of MSG mRNA

All primers used in this study are listed in Appendix A. The expression levels of the Ser1 and Ser3 mRNAs in the MSGs were analyzed using quantitative RT-PCR. Briefly, the total RNA was isolated from homogenized MSGs of the fifth-instar larvae using an ISOGEN II RNA extraction reagent (FUJIFILM Wako Laboratory Chemicals). Each RNA sample (1 ng) was analyzed using an iTaq Universal SYBR Green One-Step Kit (Bio-Rad Laboratories, Hercules, CA, USA) with standard reagents. Ser1 and Ser3 mRNAs were quantified by PCR using the primer combinations Ser1qf and Ser1qr or Ser3qf and Ser3qr, respectively. DNA amplification was detected using a CFX96 Touch^TM^ Real-Time PCR Detection System (Bio-Rad Laboratories) with BIO RAD CFX Manager software. The relative mRNA expression levels were determined using a standard curve generated by the serial dilution of the plasmid (pGEM-T Easy) encoding the tested gene sequence. The Ser1 and Ser3 mRNAs were normalized to the level of 18S rRNA and quantified using the 18Sf and 18Sr primers.

### 4.5. Fibroin Sponge Preparation

Cocoons (1.6 g) from the Ser1-P1A269 and Ser3-P1A269 lines, as well as nontreated *w1-pnd* and fibroin fibers (0.1 g) of degummed *w1-pnd* cocoons, were immersed in 70% and 100% ethanol, air-dried, and dissolved in 12.5 mL of 9M LiBr containing 100 mM Tris-HCl (pH 9.0) for 2 h at 25 °C. Each solution was desalted in 3 L of 1 mM Tris-HCl (pH 9.0) six times using a dialysis membrane with a molecular mass cut-off of 14 kDa (size 36; FUJIFILM Wako Laboratory Chemicals). The dialysate was concentrated by placing the solution in a dialysis membrane to obtain 3 wt% fibroin solution. However, the preparation of the fibroin solution from *w1-pnd* cocoons was unsuccessful because of the remarkable loss of fibroin through precipitation during dialysis. The successfully obtained fibroin solution was mixed with 1–5% (*v*/*v*) methanol in a 35-mm dish (Iwaki), followed by storage overnight at −20 °C to facilitate the formation of porous 3D fibroin sponges. The transparency of sponge samples of the same thickness (4.0 mm) was compared by placing the samples on a black background. Photographs taken 10 cm from the sponges under light conditions were subjected to optical density measurements using ImageJ software (NIH, Bethesda, MD, USA; https://imagej.nih.gov/ij/, accessed on 11 June 2022).

### 4.6. Synchrotron WAXD Analysis

Experimental samples, such as lyophilized cocoon samples or sponges and amorphous samples prepared by the lyophilization of the fibroin solutions described above (Section 4.3), were cut into pieces and subjected to WAXD measurements at room temperature. Measurements were performed at beamline BL-6A of the Photon Factory, a synchrotron radiation facility at the High-Energy Accelerator Research Organization (KEK) in Tsukuba, Japan. This beamline utilized a bent plane mirror (composed of ultra-low expansion glass) and a curved monochromator (of a triangular germanium (111) crystal) to focus the X-ray beam at a distance of 2.5 m from the sample position. The beam size at the focal position was approximately 0.50 mm (horizontal) and 0.25 mm (vertical) (in terms of HWHM values). Since the distance from the sample to the detector (an imaging plate, as described below) position was set at 10 cm, the incident beam at the detector position was slightly off-focus. Details on the beamline have been reported elsewhere [31,32]. The wavelength (λ) of the incident X-ray beam was set at 0.15 nm (corresponding to 8.267 keV of photon energy). The magnitude of the scattering vector *q* was calibrated using a polyethylene crystal. Here, *q* is defined as ǀ*q*ǀ = *q* = (4π/λ)sin(*θ*/2), where *θ* is the scattering angle. The WAXD pattern was recorded for an X-ray exposure time of 10 s using an imaging plate (BAS-IP MS 2025, Fuji Photo Film, Tokyo, Japan; size: 200 × 250 mm^2^) with an actual pixel size of 100 × 100 μm^2^. BAS2500 (Fuji Photo Film) was used to develop the exposed images on the imaging plate. Background scattering was subtracted by considering the transmission values of the samples. One-dimensional WAXD profiles were obtained by calculating the circular average of the two-dimensional WAXD pattern. Thus, the obtained WAXD profile was further normalized by averaging the scattering intensity. To evaluate the degree of crystallinity in the samples, the WAXD profiles were subjected to peak deconvolution (including crystalline and amorphous peaks) using homemade software by assuming the Lorentz function for the crystalline reflection peak and by adjusting the fraction of amorphous halo scattering (assuming that the WAXD profile for amorphous fibroin was included while maintaining the same shape of the profile, but partially according to the amorphous component in the normalized WAXD profiles of the other crystalline samples, as shown by the blue broken curves in Appendix A).

### 4.7. Cultivation of ATDC5 Cells on Fibroin Coats

The obtained fibroin solution (1 mL) was spread evenly in 24-well culture plates (Iwaki). The solution was removed, and ethanol (200 µL) was added, followed by storage overnight at −20 °C. Recrystallized fibroin that resembled the fibroin sponge structure was partially formed on the substrate surface. After removing the ethanol and air-drying, 2 × 10^4^ ATDC5 cells were seeded into the wells, along with 500 µL proliferation medium. A 0.1% gelatin-coated well was used as a control. The proliferation of ATDC5 cells was monitored using the Cell Counting Kit-8 (Dojindo Laboratories, Kumamoto, Japan), which detects intracellular dehydrogenase activity using the water-soluble tetrazolium salt-8 (WST-8) assay. The Cell Counting Kit-8 solution (10% (*v*/*v*)) was added to each well and incubated for 2 h at 37 °C. The absorbance of each well was measured at 450 nm using an iMark microplate reader (Bio-Rad).

The chondrogenic and osteogenic differentiation of ATDC5 cells was promoted in proliferation medium supplemented with 10 µg/µL of recombinant human insulin (Roche, Basel, Switzerland) and 20 ng/mL of recombinant human BMP-2 (Proteintech, Rosemont, IL, USA) (differentiation medium). The differentiation medium in each well was changed once every two days. Cultivated cells were washed with PBS, fixed with methanol for 20 min, and incubated with Alcian blue solution for 16 h at room temperature. Cell images were captured after washing the samples with water. The stained cells were then lysed with 6 M guanidine hydrochloride solution (500 µL/well), and the absorbance of the lysates was measured at 620 nm using a microplate reader.

To detect mineralization, cells cultured for 15 or 28 days were fixed in methanol for 20 min. The cells were then stained with alizarin red solution for 16 h at room temperature before capturing the cell images. The cells were lysed in 20% formic acid solution (500 µL/well), and the absorbance of the lysates was measured at 450 nm using a microplate reader.

### 4.8. Statistical Analysis

Each assay was performed several times (the number of replicates is indicated in the figure legends as *n*). The mean ± standard deviation was calculated for all the experimental data, and the significance of differences between samples (*p* < 0.05) was analyzed using the Student’s *t*-test for two-sample comparisons or one-way analysis of variance (ANOVA) followed by Tukey’s test for pairwise multiple comparisons.

## 5. Conclusions

We generated Ser1-P1A269 and Ser1/3-P1A269 transgenic silkworm lines that produced cocoons composed solely of fibroin, eliminating the need for laborious filatures and degumming steps to remove sericin from conventional cocoons and reduce allergen risks. Recent developments have also provided methods for producing regenerated fibroin materials with higher biocompatibility.

## Figures and Tables

**Figure 1 ijms-23-07433-f001:**
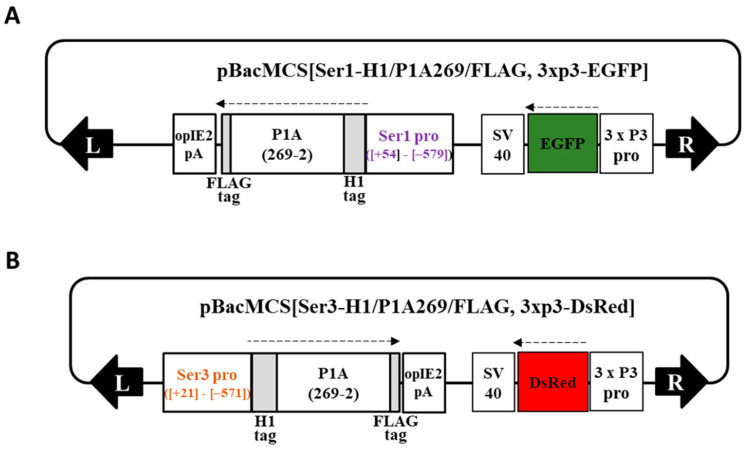
Schematic representation of the donor plasmids. (**A**) pBacMCS[Ser1-H1/P1A269/FLAG, 3 x P3-EGFP] and (**B**) pBacMCS[Ser3-H1/P1A269/FLAG, 3 x P3-DsRed] containing designed sequences expressing the pierisin-1A (P1A) catalytic domain under the control of the Ser1 and Ser3 promoters, respectively, along with the genetic markers. The numbers indicate the nucleotide position of the gene promoter sequence [29,30] and 2–269 amino acid positions of P1A [20]. Black arrows indicate the *piggyBac* inverted terminal repeat (ITR) at the left and right arms; broken arrows indicate the direction of the translation.

**Figure 2 ijms-23-07433-f002:**
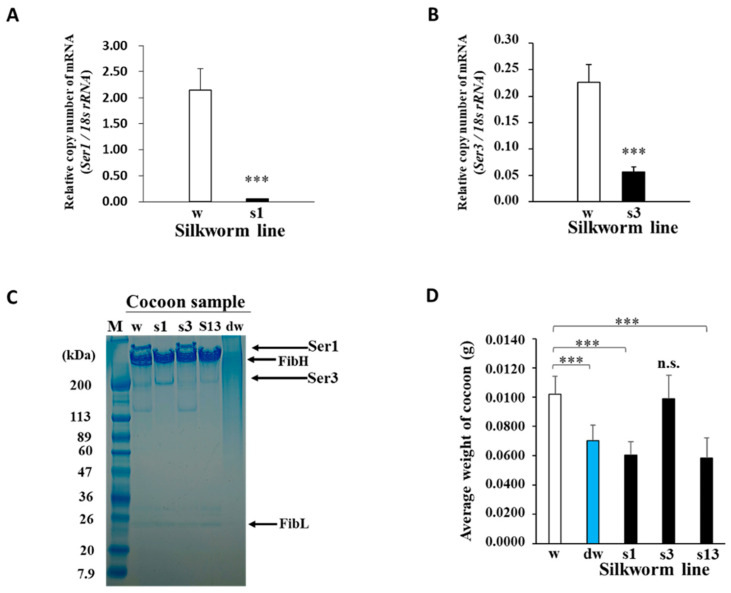
Functional effects of P1A269 expression in middle silk glands (MSGs). (**A**,**B**) Quantitative RT-PCR analysis of the relative Ser1 (s1) and Ser3 (s3) mRNA levels in the Ser1-P1A269 and Ser3-P1A269 lines, respectively. Total RNAs were isolated from the MSGs of the respective larvae. The mRNA levels of Ser1 at day 5 of the fifth instar of the Ser1-P1A269 line and Ser3 at day 6 of the fifth instar of the Ser3-P1A269 line were quantified and compared with that of *w1-pnd* (w) at the same developmental stage, respectively. The expression levels of Ser1 and Ser3 mRNAs were normalized to that of 18S ribosomal RNA. Data are presented as the means ± SD of relative values (copy number); *n* = five (for Ser1 mRNA) and *n* = eight (for Ser3 mRNA) independent samples; *** *p* < 0.001 versus *w1-pnd* (Student’s *t*-test). (**C**) SDS–PAGE analysis of the cocoons produced by *w1-pnd* (lane w), the Ser1-P1A269 line (lane s1), Ser3-P1A269 (lane s3), Ser1/3-P1A269 (lane s13), and degummed *w1-pnd* (lane dw). The cocoons were dissolved in concentrated LiBr solution, diluted in water, and electrophoresed on a 5–20% gradient gel, along with the protein marker (lane M). Arrows on the right indicate the positions of FibH, FibL, Ser1, and Ser3 [4,5]. (**D**) Comparison of the cocoon weights. The dry weight of *w1-pnd* (w), degummed *w1-pnd* (dw), the Ser1-P1A269 line (s1), Ser3-P1A269 (s3), and Ser1/3-P1A269 (s13) cocoons were measured. Data are presented as the means ± SD; *n* = 50 independent samples. *** *p* < 0.001 (one-way ANOVA, followed by Tukey’s test); n.s., not significant versus *w1-pnd*.

**Figure 3 ijms-23-07433-f003:**
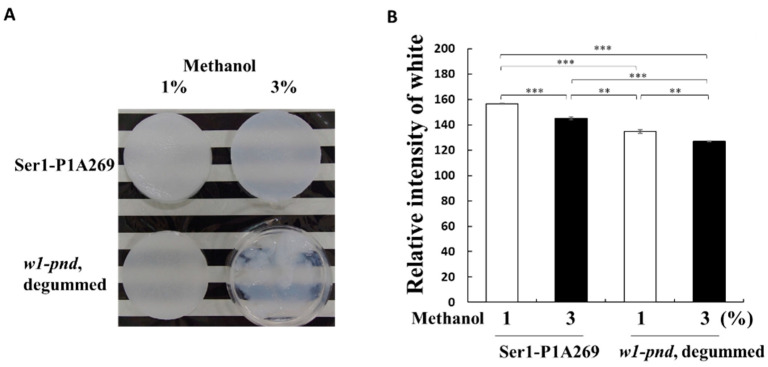
Superiority of cocoons with a high fibroin content for the formation of fibroin sponges. (**A**) Comparison of the appearance of sponges of degummed *w1-pnd* and non-degummed Ser1-P1A cocoons formed with 1% or 3% (*v*/*v*) methanol. (**B**) Transparency analysis of sponges prepared from the Ser1-P1A269 cocoon and degummed *w1-pnd* by using 1% and 3% (*v*/*v*) methanol. Transparency of the sponges was measured using computer-assisted image analysis software. Data are the means ± SD of relative intensity; *n* = three independent samples. ** *p* < 0.01 and *** *p* < 0.001 (one-way ANOVA, followed by Tukey’s test).

**Figure 4 ijms-23-07433-f004:**
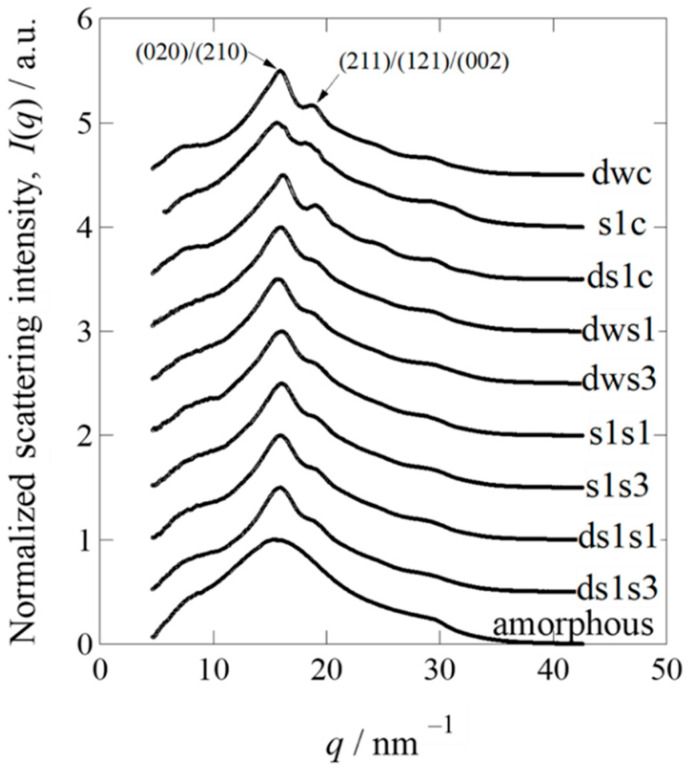
Normalized one-dimensional wide-angle X-ray diffraction profiles of degummed *w1-pnd* (dwc), Ser1-P1A269 (s1c), and degummed Ser1-P1A269 (ds1c) cocoons, as well as sponges prepared from degummed *w1-pnd* with 1% (dws1) and 3% (dws3) methanol, Ser1-P1A269 with 1% (s1s1) and 3% (s1s3) methanol, and degummed Ser1-P1A269 with 1% (ds1s1) and 3% (ds1s3) methanol. To avoid overlap, all profiles were vertically shifted at 0.5 intervals, except for the amorphous sample (no vertical shift).

**Figure 5 ijms-23-07433-f005:**
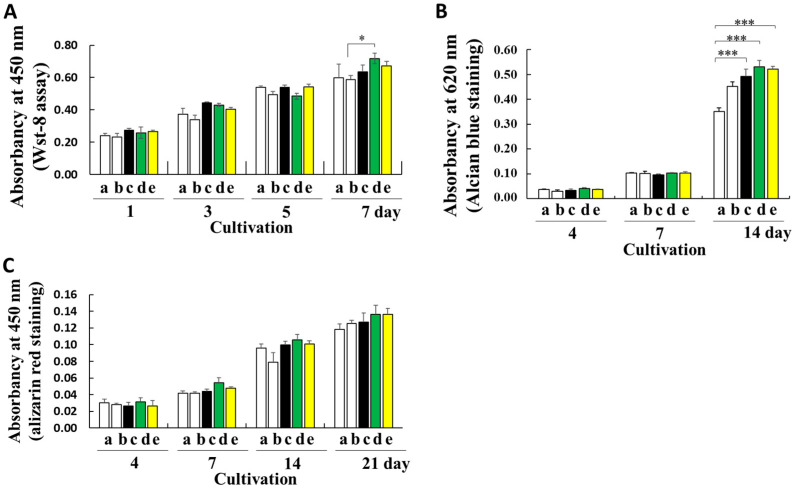
Cultivation of ATDC5 cells on fibroin biomaterials. (**A**) Comparison of cell viability on noncoated substrates (column a), gelatin-coated substrates (column b), and substrates coated with fibroin solutions prepared from degummed *w1-pnd* (column c), non-degummed Ser-P1A269 (column d), and Ser1/3-P1A269 (column e) cocoons by measuring the WST-8 reaction on the indicated day after cell seeding. Data are shown as the means ± SD; *n* = four independent samples; * *p* < 0.05 (one-way ANOVA, followed by Tukey’s test). (**B**) Early chondrogenic differentiation of ATDC5. The ATDC5 cells were cultured in a differentiation medium; on the indicated day, signs of early differentiation such as cartilage nodule formation were investigated by measuring the amount of glycosaminoglycan produced via Alcian blue staining. Data are shown as the means ± SD; *n* = four independent samples; *** *p* < 0.001 (one-way ANOVA, followed by Tukey’s test). (**C**) Late osteogenic differentiation of ATDC5 cells. During the late osteogenic differentiation of ATDC5, the mineralization of the extracellular matrix was assessed by alizarin red staining. Data are shown as the means ± SD; *n* = four independent samples (not significant by one-way ANOVA with Tukey’s test).

**Table 1 ijms-23-07433-t001:** Wide-angle X-ray diffraction analysis of the cocoons and fibroin sponges.

		(020)/(210) **		(211)/(121)/(002)	

Samples	Crystallinity	Peak Position/nm^−1^	HWHM/nm^−1^	Peak Position/nm^−1^	HWHM/nm^−1^
dwc	0.32 *	15.67	1.54	18.71	0.81
s1c	0.26	15.45	1.83	19.15	0.59
ds1c	0.26	15.94	1.39	19.00	0.50
dws1	0.28	15.82	1.59	19.33	1.18
dws3	0.24	15.60	1.53	19.13	1.03
s1s1	0.27	15.87	1.71	19.45	1.45
s1s3	0.24	15.84	1.51	19.36	1.40
ds1s1	0.28	15.94	1.61	19.45	1.33
ds1s1	0.25	15.79	1.56	19.31	1.40

* Data are shown in approximate number to the second decimal place. ** Peak position.

## Data Availability

The data presented in this study are available upon request from the authors.

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
