# Peer review of "Bioengineered Silkworm for Producing Cocoons with High Fibroin Content for Regenerated Fibroin Biomaterial-Based Applications"

_ijms, 2022, doi:10.3390/ijms23137433_

Round 1

Reviewer 1 Report

The author used bioengineered silkworms to produce the sericine free cocoons. In transgenic silkworms, sericin expression in the middle silk glands is suppressed during the late larval stage by ectopic expression of the cabbage butterfly-derived cytotoxin pierisin-1A. The effect of expression of this cytotoxin driven by sericine 1 and sericine 3 promoters, respectively, on production of sericine 1 and sericine 3 were evaluated by both quantitative RT-PCR and SDS-PAGE. The authors further studied the properties of the transgenic cocoons. Transgenic cocoons have low crystallinity (detected by X-ray analysis) and the application of alcohol allows the formation of fibroin sponges.

This study represents a major step in the efficient production of biomaterials as there is no need to subsequently remove sericin from the cocoons. The research topic is well introduced, all methods are well described, and results are clearly presented. But there is two crucial thing missing from the article:

    1) I think that readers would be interested to know how the cocoons of these transgenic silkworm differ from cocoons of control animals. Do you have any photo documentation i.e. either macrophotography (if the difference is significant at this level of resolution) or scanning electron microscope images? The reduction of sericine level shout be obvious especially in the inner surface of the cocoon. I truly believe that including photodocumentation will increase the quality of the manuscript.

22) It would also be useful to measure the mechanical properties, especially the  strength of the fibers produced by these transgenic silkworms. 

Minor comments:

·       Supplementary figure 1 - w1-pnd line (W), there is a lower-case letter for w – in the image, please specify what the arrow point to

·       Supplementary figure 4 - some numbers in the graphical representation cannot be read, write them next to the columns. Also, the shades of colors are very similar and cannot be distinguished well.

·       Supplementary figure 7 - two of graphical analyses are merged into one image – please correct it. Also, please, use the same font as in the Figure S2 and S3   

·       Reference no.7 – please, correct the word “characteristics”

Reviewer 2 Report

The manuscript is " Bioengineered Silkworm for Producing Cocoons with High Fi- 2 broin Content for Regenerated Fibroin Biomaterial-Based Ap- 3 plications ".

General comments:

This study to explore the transgenic silkworms produce high fibroin content for biomaterial application. However, there is still some work to be done.

1.     Abbreviations should be clearly presented.

2.     The biomedical application is only presented in Figure 5, which is not enough to fully express its advantages, and more data should be added.

3.     This article is not too long, and some attachment data can be added to the main text.

4.     The article does not show the stability and longevity of the transgenic gene.

5.     The structure and characteristics of biomaterial fabricated from the transgenic cocoons are not clear.
